# The Association of suPAR with Cardiovascular Risk Factors in Young and Healthy Adults

**DOI:** 10.3390/diagnostics13182938

**Published:** 2023-09-13

**Authors:** Niklas Fabio Wohlwend, Kirsten Grossmann, Stefanie Aeschbacher, Ornella C. Weideli, Julia Telser, Martin Risch, David Conen, Lorenz Risch

**Affiliations:** 1Faculty of Medicine, University of Bern, 3012 Bern, Switzerland; 2Dr. Risch Medical Laboratory, 9490 Vaduz, Liechtenstein; 3Faculty of Medical Sciences, Private University in the Principality of Liechtenstein, 9495 Triesen, Liechtenstein; 4Cardiovascular Research Institute Basel, Division Cardiology, University Hospital Basel, University of Basel, 4031 Basel, Switzerland; 5Soneva Fushi, Boduthakurufaanu Magu, Male 20077, Maldives; 6Division of Laboratory Medicine, Cantonal Hospital Graubünden, 7007 Chur, Switzerland; 7Population Health Research Institute, McMaster University, Hamilton, ON L8L 2X2, Canada; 8Department of Laboratory Medicine, Institute of Clinical Chemistry, Inselspital Bern University Hospital, University of Berne, 3012 Berne, Switzerland

**Keywords:** soluble urokinase plasminogen activator receptor (suPAR), biomarkers, cardiovascular risk, prevention

## Abstract

The soluble urokinase plasminogen activator receptor (suPAR), as a correlate of chronic low-grade inflammation, may be used to predict individual cardiovascular risk. Since chronic low-grade inflammation is thought to be associated with the development of cardiovascular disease, this study aimed to evaluate if suPAR plasma levels are correlated with cardiovascular risk factors in young and healthy adults (aged 25–41 years). Consequently, data from the GAPP (genetic and phenotypic determinants of blood pressure and other cardiovascular risk factors) study were used to investigate suPAR plasma levels in relation to the following cardiovascular risk factors and laboratory parameters: BMI, physical activity, alcohol consumption, smoking status, blood pressure parameters, glucose status, and lipid levels. Additionally, suPAR was compared to the healthy lifestyle score and the Framingham score representing the overall cardiovascular risk profile. These associations were assessed using two different statistical approaches. Firstly, all cardiovascular risk factors and scores were compared amongst sex-specific suPAR plasma levels with ANOVA analysis. Secondly, sex-specific multivariable linear regressions were performed. Female participants had higher plasma suPAR levels than male participants (1.73 ng/mL versus 1.50 ng/mL; *p* < 0.001). A significant inverse correlation between suPAR plasma levels and HDL cholesterol was found in men (*p* = 0.001) and women (*p* < 0.001). Furthermore, male (*p* < 0.001) and female participants (*p* < 0.001) who smoked showed significantly higher plasma levels of suPAR (*p* < 0.001). For male participants, an inverse correlation of the healthy lifestyle score with suPAR plasma levels (*p* = 0.001) and a positive correlation of the Framingham score with suPAR plasma levels (*p* < 0.001) were detected. In women, no such correlation was found. The cholesterol levels (*p* = 0.001) and HbA1c (*p* = 0.008) correlated significantly with plasma suPAR levels in female participants. suPAR plasma levels were found to be strongly associated with certain cardiovascular risk factors; however, sex-specific differences were found. These sex-specific differences might be explained by the higher prevalence of cardiovascular risk factors in men resulting in a stronger correlation of suPAR as a marker of low-grade inflammation, since the existence of the risk factors already led to subclinical damage in men. Further research on suPAR levels in an older study population is needed.

## 1. Introduction

As a result of the globally increasing prevalence of cardiovascular risk factors and cardiovascular diseases, research nowadays not only focuses on possible causes and appropriate treatments, but furthermore, on possible preventive measures as well [1,2]. Cardiac biomarkers are consequently being investigated as possible components of personalized risk stratification strategies to customize appropriate preventive measures individually [3].

Compared with the fluctuating plasma levels of hsCRP, the plasma levels of the soluble urokinase plasminogen activator receptor (suPAR) are regarded as being more stable [4]. Therefore, suPAR is being examined as a potential alternative biomarker and a more accurate estimate of the inflammation status [4]. The soluble urokinase plasminogen activator receptor is produced as a cleavage product of the membrane-bound uPAR [5]. This receptor is activated by uPA and its signaling pathway regulates the coagulation cascade and cell signaling, influencing the survival and proliferation as well as the motility of cells [6]. uPAR is located on a wide variety of cells including immune cells and structural cells (such as keratinocytes, fibroblasts, as well as endothelial cells), but also on megakaryocytes, smooth muscle cells, and certain tumor cells [7]. Inflammation is thought to upregulate uPAR expression [8].

The primary aim of the study was to evaluate the association of suPAR plasma levels with cardiovascular risk factors. Additionally, the association of suPAR plasma levels with the overall cardiovascular risk estimated by the healthy lifestyle score and the Framingham risk score was assessed [9,10]. For the verification of predicted correlations between plasma levels of suPAR and cardiovascular risk factors in younger populations, the current study analyzed data from young and healthy adults aged 25–41 years.

## 2. Methods

### 2.1. Study Population

This substudy was based on the study population of the GAPP (genetic and phenotypic determinants of blood pressure and other cardiovascular risk factors) study. The GAPP project was constructed as a cohort study including the population of the Principality of Liechtenstein [11]. The overall aim of the GAPP study is to determine the possible causes and risk factors of arterial hypertension and cardiovascular disease. Exclusion criteria for participation were pre-existing cardiovascular disease or pre-existing documented obstructive sleep apnea syndrome and obesity class II, pregnancy or ongoing lactation, the daily intake of medication (such as antidiabetic drugs or insulin, as well as nonsteroidal anti-inflammatory drugs including aspirin or steroids (>1 day per week) and sympathomimetic drugs (>1 day per week)), the frequent consumption of liquorice (>1 day per week), or any known severe diseases [11]. Of the 2170 participants enrolled at baseline, 134 participants were excluded due to missing or invalid suPAR values (*n* = 66), the intake of antihypertensive treatment (*n* = 34), levels of creatinine >300 umol/L (*n* = 1), or missing parameters that are used as correction factors (*n* = 33), resulting in 2036 participants becoming eligible for the overall analyses (Figure 1). A subpopulation excluded participants with missing ambulatory blood pressure measurements (*n* = 63), or with fewer than 10 daytime or fewer than 5 nighttime blood pressure measurements (*n* = 22), resulting in 1951 participants (Figure 1). Written informed consent was obtained and the local ethics committee (KEK, Zürich, Switzerland) approved the study protocol.

### 2.2. Assessment of Laboratory Parameters

Laboratory parameters were acquired through a fasting venous blood sample taken via a minimally invasive venipuncture and a morning urine sample. The blood samples were immediately analyzed in an accredited medical laboratory (ISO 17025) and parts of them were stored at −80 °C directly after centrifugation for possible future analyses [12]. EDTA plasma samples were used for the determination of plasma levels of suPAR using the enzyme immunoassay (suPARnostic, ViroGates, 3460 Birkerød, Denmark). The kidney function was estimated using the Chronic Kidney Disease Epidemiology Collaboration (CKD-EPI) equation [13]. Additionally, urinary sodium and creatinine levels were determined. A complete list of all laboratory parameters was previously described in the first paper published of the GAPP study project [11].

### 2.3. Cardiovascular Risk Profile Algorithms

We used 7 predefined health metrics as indicators for a healthy lifestyle summed up in a healthy lifestyle score [9,14]. If one of the health metrics was fulfilled, 1 point was added to the total score ranging from a minimum of 0 points to a maximum of 7 points. These seven health metrics included in the healthy lifestyle score were: physical activity exceeding or equaling 150 min of intense physical activity or 210 min of moderate physical activity per week, a BMI lower than 25 kg/m^2^, a nonsmoker status, an LDL plasma level of less than 160 mg/dL, an HbA1c of less than 5.7%, a systolic blood pressure < 120 mmHg, and healthy nutrition. The participants’ nutrition was qualified as healthy if at least two of the following criteria were met: the consumption of at least 4 portions of fruit or vegetables in a day, the consumption of fish at least 2 times a week, and a daily sodium intake lower than 1.5 g. Daily sodium intake was estimated using the Kawasaki formula, which calculates an estimate of 24 h sodium excretion based on natrium and creatinine in spot urine [15].

In addition, the Framingham score was used as an indicator of the individual cardiovascular risk profile. The score was based on each participants’ age, the HDL serum levels, the total cholesterol serum levels, the systolic blood pressure, the smoking status, and the glycemic status [10]. 

### 2.4. Blood Pressure Measurements

The conventional office blood pressure (BP) was measured three times on the nondominant arm using a validated oscillometric blood pressure monitor (Microlife BP3AG1, Microlife AG, Widnau, Switzerland). The mean of the conventional BP was calculated from the second and third blood pressure measurement. The mean arterial pressure (MAP) was calculated using an alternative equation, which contrasts the conventional formula by using the form factor 0.412 instead of 0.333 [16,17]. The MAP estimate calculated by this alternative equation seems to be in a closer relation with cardiovascular parameters (such as left ventricular mass, aortic stiffness, and carotid wall thickness) than the conventional one [16]. The MAP for the second and third office BP measurement was calculated and the mean of these two calculated MAPs was used for the statistical analyses. In this study, a systolic pressure of 140 mmHg or higher and a diastolic pressure of 90 mmHg were qualified for the diagnosis of arterial hypertension [18].

The 24 h blood pressure was recorded every 15 min from 07:30 to 22:00 and every 30 min from 22:00 to 07:30 with a validated ambulatory blood pressure monitoring system (BR-102 plus, Schiller AG, Baar, Switzerland). The participants were requested to keep their arm still during the time of recording, but were allowed to participate normally in their daily routine. The differentiation between day- and nighttime measurements was assigned individually according to the participants’ sleep diary kept during the measurement periods. The measurement of the 24 h BP was repeated if more than 20% of the BP measurements were invalid.

### 2.5. Statistical Analysis

Since this study confirmed sex-specific differences in mean suPAR plasma levels, the baseline characteristics, as listed in Table 1, describe the female and male study population separately and all of the following analyses were carried out sex-specifically [19]. All investigated cardiovascular risk factors, consisting of the BMI (kg/m^2^), an estimate for the physical activity, alcohol drinking and smoking habits, blood pressure indicators, and laboratory parameters (describing the glycemic profile, the blood lipids, renal function and hsCRP) that complemented the cardiovascular risk estimation scores or were used as correction factors, were represented in the sex-specific baseline characteristics. The distribution pattern of continuous variables was visually analyzed. Normally distributed variables are presented as means and were analyzed using the *t*-test. Atypically distributed variables are presented as medians and were analyzed using the Wilcoxon test. Categorical variables are presented in percentages and were analyzed using the chi-square test or Fisher’s exact test as appropriate.

In order to confirm the results, two different statistical approaches were used. Firstly, cardiovascular risk factors were compared between sex-specific quartiles of suPAR plasma levels using one-way ANOVA tests.

In a second step, we performed multivariable linear regression analyses of each cardiovascular risk factor continuously and also across sex-specific quartiles of suPAR plasma levels. The following variables were used as correction factors for the multivariable linear regression analyses: age, BMI, physical activity, HbA1c, renal function estimated by GFR, LDL, HDL, smoking status, and hsCRP. For the multivariable linear regressions analyzing the correlation of the healthy lifestyle score and the Framingham score with suPAR plasma levels, only age and eGFR were used as correction factors.

We considered correlations to be confirmed if they were proven in these two mentioned statistical approaches.

The statistical analyses were performed with RStudio (2021.09.1.372, Posit, Boston, MA, USA).

## 3. Results

Baseline characteristics stratified by sex are presented in Table 1. While there was no difference in age and hsCRP between the male and female study population, sex-specific differences were found for suPAR plasma levels (1.50 ng/mL vs. 1.73 ng/mL, *p* < 0.001) and all cardiovascular risk factors were investigated.

When comparing the means and medians of the cardiovascular risk factors between the quartiles of suPAR plasma levels in the male study population as listed in Table 2, positive correlations were found for BMI (*p* = 0.014), a comparison between current and past smoking statuses and those of nonsmokers (*p* < 0.001), cholesterol (*p* = 0.034), HDL cholesterol (*p* < 0.001), as well as an inverse correlation for HDL (*p* < 0.001). The healthy lifestyle score was inversely (*p* < 0.001) correlated with suPAR plasma levels and the Framingham score (*p* < 0.001) was positively correlated with suPAR plasma levels.

These correlations were confirmed using the multivariable regression analyses, as presented in Table 3 for the variables HDL with a standardized β-regression coefficient of −0.155 (*p* < 0.001) and current smoking compared to past and completely absent smoking habits with a standardized β-regression coefficient of 0.267 (*p* < 0.001). An additional significant correlation was found for the variable irregular physical activity with a standardized β-regression coefficient of 0.780 (*p* = 0.014). Multivariable regression analyses also confirmed the correlation of suPAR plasma levels with cardiovascular risk profile algorithms, the healthy lifestyle score with a standardized β-regression coefficient of −0.129 (*p* = 0.001), and the Framingham score with a standardized β-regression coefficient of 0.161 (*p* < 0.001).

Table 4 shows positive correlations in the female population of suPAR plasma levels with the following risk factors: BMI (*p* < 0.001), a comparison between current and past smoking statuses and those of nonsmokers (*p* < 0.001), fasting glucose (*p* < 0.001), HbA1c (*p* < 0.001), cholesterol (*p* = 0.001), and an inverse correlation for HDL (*p* < 0.001). An inverse correlation was found between the suPAR plasma levels and the healthy lifestyle score (*p* = 0.005).

When assessing correlations in the female study population using multivariable regression analyses as listed in Table 5, a confirmation of the earlier-mentioned correlations was found for HDL with a standardized β-regression coefficient of –0.114 (*p* < 0.001), HbA1c with a standardized β-regression coefficient of 0.081 (*p* = 0.008), cholesterol with a standardized β-regression coefficient of −0.094 (*p* < 0.001), and current smoking compared to past and completely absent smoking habits with a standardized β-regression coefficient of 0.076 (*p* = 0.017). The correlation of suPAR plasma levels and the healthy lifestyle score was not confirmed using multivariable regression analyses.

## 4. Discussion

This study showed significant higher suPAR plasma levels in the female population compared to the male population and sex-specific differences in the distribution of cardiovascular risk factors [20,21].

In accordance with recent studies, a statistically significant inverse correlation between suPAR plasma levels and the HDL serum levels was found in the male and female population, as well as higher suPAR plasma levels in smokers compared to nonsmokers and past smokers [21,22]. Furthermore, our study confirmed a positive correlation between cholesterol levels and suPAR plasma levels in the female population [21]. The biomarker suPAR is regarded as a marker of endothelial dysfunction and therefore also of atherosclerosis, which is why it seems plausible that smoking and higher cholesterol levels increase, whereas a higher HDL decreases plasma suPAR levels [23].

In contrast to other studies, our study could not confirm a correlation between BMI and suPAR plasma levels [21]. The highest BMIs observed in the current study population (BMI of 38 kg/m^2^) are not classified as obesity grade III (BMI ≥ 40 kg/m^2^), which was associated with higher suPAR plasma levels in the study published by Haupt, T.H. et al. [21]. The MONICA study resulted in the same conclusion as our study: that BMI is not independently associated with suPAR plasma levels [24]. This conclusion could imply that comorbidities associated with obesity are causing higher plasma levels of suPAR, rather than a higher BMI itself.

To the existing literature with an inconsistent evaluation of the association of suPAR and diabetes mellitus type 2, we now add that higher HbA1c in the female population seems to be correlated with higher suPAR plasma levels [21,24]. The sex-specific difference in this correlation cannot be explained by the already-existing literature, since a sex-independent correlation of suPAR and the endothelial dysfunction caused by permanently higher blood sugar levels reflected by the HbA1c levels would be expected [23,25].

Additionally, we used the healthy lifestyle score and the Framingham score as markers of the overall cardiovascular risk in our analyses, which correlated with suPAR plasma levels only in the male study population, but not in the female one. It was shown that in male participants, a higher lifestyle score, indicating a healthier way of living, resulted in statistically lower suPAR plasma levels, while a higher Framingham score, indicating a higher cardiovascular risk and mortality, was correlated with higher suPAR plasma levels [21].

Most research attempting to assess suPAR as an estimate for cardiovascular risk is based on study populations with a distinctly higher average age of around 50 years. While these studies predict correlations between plasma suPAR levels and the participants cardiovascular risk, our study results did not detect sex-independent consistency [21,22,24]. This is why we think that in elderly female participants with a higher prevalence of cardiovascular risk factors, such correlations might be found too. This suggests that suPAR might act as a predictor of the cardiovascular risk in elderly participants.

We did not find sex-independent correlations apart from the inverse correlation of plasma suPAR levels with HDL levels and the higher suPAR plasma levels found in smokers compared to nonsmokers and current smokers [26]. The consistency of these two correlations arguably displays the extent of the influence of smoking and lower HDL levels on cardiovascular health.

Nevertheless, there is an evident indication of a certain correlation of plasma suPAR levels with cardiovascular risk factors given statistically significant correlations of the healthy lifestyle score and the Framingham score with suPAR plasma levels in the male study population. Sex-specific differences in the correlation of suPAR plasma levels with these two validated scores might be a consequence of the higher prevalence of cardiovascular risk factors in men compared to women, as demonstrated in the sex-specific baseline characteristics [27]. An increase in cardiovascular risk factors in older women might explain age-specific correlations in women, which were demonstrated in other studies on this topic. This seems quite conceivable since this biomarker represents low-grade inflammation and only the persistent influence of cardiovascular risk factors results in accumulating damage, which is then detectable later on [28].

## 5. Strengths and Limitations

A major strength of our study is the young and healthy study population (aged 25–41 years) lacking any relevant comorbidities, thus minimizing possible interference by unknown confounders. Investigating the correlations of the biomarker suPAR with cardiovascular risk factors and the overall cardiovascular risk in a young and healthy population is important, since preventive measures to reduce the global burden of cardiovascular risk factors and diseases should take place before irreversible damage is set. Therefore, potential biomarkers should be well investigated in younger populations without manifestations of cardiovascular diseases to enable an early implementation of these biomarkers.

Furthermore, the great number of variables representing the cardiovascular risk factors results in a detailed description of the study population, which allows differentiated results.

Analyzing correlations in two different ways, firstly by comparing the quartiles and secondly using multivariable linear regression analyses, ensures reliable results.

A limitation of this study is the reduction in the size of the study population due to sex-specific analyses. These separations were necessary due to the statistically significant sex-specific differences in plasma levels of suPAR and in most of the investigated variables. Secondly, the generalizability of this study is limited, given the fact that the study only included people living in Liechtenstein.

Thirdly, this study was based on investigating possible correlations of the potential biomarker suPAR with cardiovascular risk factors. The insights gained by this study should be extended by integrating the hard clinical endpoints such as survival, mortality, and disease-free survival in future follow-up studies.

In conclusion, the evident indication that there is a certain correlation between suPAR plasma levels and cardiovascular risk factors, as well as the fact that there are suspected age-specific differences in these correlations, emphasizes the importance of further investigations in this topic and particularly of comparing the mentioned potential correlations in different age groups. In our opinion, a possible approach for further investigations would be a follow-up study of equal study design, hypothesis, and the same study population, with the only difference being an older study population, in order to evaluate if suPAR might be an age-specific biomarker reflecting cardiovascular risk solely in an elderly population. A second step would be to correlate individual changes in the plasma suPAR levels of each participant with the change in their individual cardiovascular risk profile. 

## Figures and Tables

**Figure 1 diagnostics-13-02938-f001:**
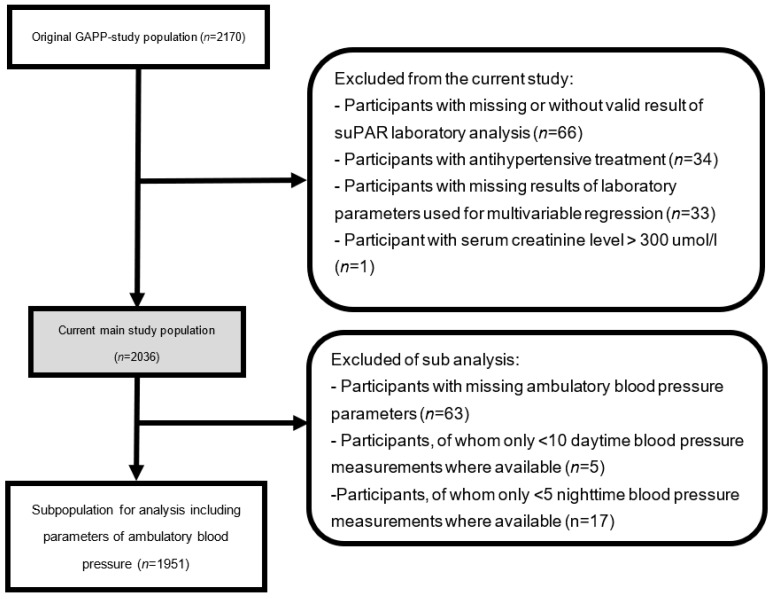
Study population.

**Table 1 diagnostics-13-02938-t001:** Baseline characteristics.

Total Study Population	*n* = 2036		
Sex	Men (*n* = 929)	Women (*n* = 1107)	
Age, years	37 (31–40)	37 (31–40)	0.667
BMI, kg/m^2^	25.8 ± 3.3	23.4 ± 3.8	<0.001
Regular physical activity, *n* (%)	683 (74%)	733 (66%)	<0.001
Irregular physical activity, *n* (%)	246 (26%)	374 (34%)	
Alcohol consumption, g/day	1.4 (0.0–3.0)	0.0 (0.0–0.8)	<0.001
Smoking (current), *n* (%)	237 (26%)	207 (19%)	<0.001
Smoking (past), *n* (%)	222 (24%)	251 (23%)	
Smoking (never), *n* (%)	470 (50%)	649 (58%)	
Mean office SBP, mmHg	127 ± 11	113 ± 10	<0.001
Mean office DBP, mmHg	82 ± 8	75 ± 8	<0.001
Mean arterial pressure, mmHg	101 ± 9	91 ± 8	<0.001
Fasting glucose, mmol/L	5.0 ± 0.5	4.7 ± 0.4	<0.001
HbA1c, %	5.4 ± 0.4	5.4 ± 0.4	0.013
Cholesterol, mmol/L	5.1 ± 1.0	4.8 ± 0.8	<0.001
LDL-C, mmol/L	3.3 ± 0.9	2.7 ± 0.7	<0.001
HDL-C, mmol/L	1.3 ± 0.3	1.7 ± 0.4	<0.001
suPAR, ng/mL	1.5 ± 0.6	1.7 ± 0.1	<0.001
hsCRP, mg/L	1.70± 3.8	1.97 ± 4.7	0.152
eGFR, mL/min/1.73 m^2^	116 ± 14	140 ± 18	<0.001
Healthy lifestyle score	4 ± 1.2	5 ± 1.1	<0.001
Framingham score	2 ± 2	3 ± 1.4	<0.001
Subgroup (*n* = 1951)	Subgroup (*n* = 897)	Subgroup (*n* = 1054)	
Mean 24 h SBP, mmHg	130 ± 10	117 ± 9	<0.001
Mean daytime SBP, mmHg	134 ± 11	121 ± 9	<0.001
Mean nighttime SBP, mmHg	114 ± 11	104 ± 10	<0.001
Mean 24 h DBP, mmHg	82 ± 8	75 ± 7	<0.001
Mean daytime DBP, mmHg	85 ± 8	79 ± 7	<0.001
Mean nighttime DBP, mmHg	69 ± 8	64 ± 7	<0.001

BMI = body mass index; SBP = systolic blood pressure; DBP = diastolic blood pressure; 24 h BP = 24 h blood pressure; BP = blood pressure; HbA1c = glycated hemoglobin 1Ac; LDL-C = low-density lipoprotein cholesterol; HDL-C = high-density lipoprotein cholesterol; suPAR = soluble urokinase plasminogen activator receptor; hsCRP = high-sensitivity C-reactive protein; eGFR = glomerular filtration rate.

**Table 2 diagnostics-13-02938-t002:** Baseline characteristics of the male study population according to suPAR quartiles.

Male Study Population	*n* = 929				
Quartiles Based on suPAR Levels	Quartile 1 (*n* = 233)	Quartile 2 (*n* = 232)	Quartile 3 (*n* = 232)	Quartile 4 (*n* = 232)	*p*-Value
suPAR Range (ng/L), men	≤1.10	1.10–1.40	1.40–1.70	≥1.70	
Age, years	37 (32–41)	36 (30–40)	37 (32–40)	36 (31–40)	0.851
BMI, kg/m^2^	25.5 ± 2.8	25.6 ± 3.2	26.1 ± 3.4	26.1 ± 3.6	0.014
Regular physical activity, min/week	168 (72%)	159 (69%)	175 (75%)	181 (78%)	0.257
Irregular physical activity, min/week	65 (28%)	73 (31%)	57 (25%)	51 (22%)	
Alcohol consumption, g/day	1.7 (0.6–3.0)	1.4 (0.0–3.1)	1.4 (0.0–2.5)	0.9 (0.0–2.4)	0.329
Smoking (current), *n* (%)	46 (20%)	43 (18%)	47 (20%)	101 (44%)	
Smoking (past), *n* (%)	51 (22%)	60 (26%)	55 (24%)	56(24%)	<0.001
Smoking (never), *n* (%)	136 (58%)	129 (56%)	130 (56%)	75 (32%)	
Mean office SBP, mmHg	127 ± 10	128 ± 11	128 ± 12	127 ± 10.9	0.402
Mean office SBP, mmHg	82 ± 8	82 ± 8	83 ± 8	82 ± 8	0.943
Mean arterial pressure, mmHg	101 ± 8	101 ± 9	101 ± 9	101 ± 9	0.632
Fasting glucose, mmol/L	5.0 ± 0.4	5.0 ± 0.4	5.0 ± 0.5	5.0 ± 0.5	0.103
HbA1c, %	5.4 ± 0.4	5.4 ± 0.4	5.4 ± 0.4	5.5 ± 0.4	0.222
Cholesterol, mmol/L	5.1 ± 0.9	5.1 ± 1.0	5.0 ± 1.0	4.9 ± 0.9	0.034
LDL-C, mmol/L	3.3 ± 0.8	3.3 ± 0.9	3.2 ± 0.9	3.2 ± 0.9	0.408
HDL-C, mmol/L	1.4 ± 0.3	1.4 ± 0.3	1.3 ± 0.3	1.2 ± 0.3	<0.001
hsCRP, mg/L	0.6 (0.4–1.2)	0.9 (0.5–1.6)	0.9 (0.5–1.9)	1.3 (0.9–2.3)	<0.001
eGFR, mL/min/1.73 m^2^	117 ± 14	117 ± 13	115 ± 14	114 ± 13	0.011
Healthy lifestyle score	3.7 ± 1.2	3.6 ± 1.3	3.6 ± 1.2	3.3 ± 1.2	<0.001
Framingham score	1.1 ± 1.8	1.4 ± 1.7	1.6 ± 1.7	1.9 ± 1.7	<0.001
Subgroup (*n* = 897)					
Quartiles based on suPAR levels	Quartile 1 (*n* = 225)	Quartile 2 (*n* = 224)	Quartile 3 (*n* = 224)	Quartile 4 (*n* = 224)	
Mean 24 h SBP, mmHg	129 ± 9	130 ± 10	130 ± 10	130 ± 10	0.727
Mean daytime SBP, mmHg	133 ± 9	134 ± 11	134 ± 11	134 ± 11	0.617
Mean nighttime SBP, mmHg	114 ± 10	114 ± 11	115 ± 11	114 ± 11	0.864
Mean 24 h DBP, mmHg	82 ± 6	82 ± 8	82 ± 8	81 ± 8	0.993
Mean daytime DBP, mmHg	85 ± 7	85 ± 9	85 ± 9	85 ± 8	0.960
Mean nighttime DBP, mmHg	69 ± 7	69 ± 8	70 ± 8	69 ± 9	0.984

suPAR = soluble urokinase plasminogen activator receptor; BMI = body mass index; SBP = systolic blood pressure; DBP = diastolic blood pressure; HbA1c = glycated hemoglobin 1Ac; LDL-C = low-density lipoprotein cholesterol; HDL-C = high-density lipoprotein cholesterol; hsCRP = high-sensitivity C-reactive protein; eGFR = glomerular filtration rate.

**Table 3 diagnostics-13-02938-t003:** Multivariable linear regression analyses for the relationship between serum levels of suPAR and blood pressure parameters in men.

Male Study Population	*n* = 929					
Quartiles Based on suPAR Levels	Continuous (*n* = 929)	Quartile 1 (*n* = 233)	Quartile 2 (*n* = 232)	Quartile 3 (*n* = 232)	Quartile 4 (*n* = 232)	*p*-Value
BMI, kg/m^2^	−0.056	reference	0.014	0.040	−0.009	0.115
Irregular physical activity, %	0.780	reference	−0.037	0.029	0.059	0.014
Alcohol consumption, g/day	0.055	reference	−0.024	−0.052	−0.055	0.085
Current smoking, %	0.267	reference	−0.107 **	−0.035	0.231 **	<0.001
Past smoking, %	0.047	reference	−0.039	0.002	0.059	0.164
Mean office SBP, mmHg	−0.020	reference	−0.011	0.054	−0.004	0.561
Mean office DBP, mmHg	0.030	reference	0.010	0.035	−0.018	0.380
Mean office MAP, mmHg	0.032	reference	≥0.001	0.049	−0.012	0.351
Fasting glucose, mmol/L	−0.003	reference	0.007	0.011	0.037	0.932
HbA1c, %	0.004	reference	−0.046	0.004	0.012	0.910
Cholesterol, mmol/L	−0.048	reference	0.001	−0.029	−0.082 *	0.149
LDL-C, mmol/L	−0.030	reference	0.023	0.001	−0.030	0.381
HDL-C, mmol/L	−0.155	reference	−0.044	−0.103 *	−0.150 *	<0.001
Healthy lifestyle score	−0.129	reference	0.035	−0.035	−0.091 **	0.001
Framingham score	0.161	reference	−0.027	0.039	0.124 **	<0.001
Subgroup (*n* = 897)						
Quartiles based on suPAR levels	Continuous (*n* = 897)	Quartile 1 (*n* = 225)	Quartile 2 (*n* = 224)	Quartile 3 (*n* = 224	Quartile 4 (*n* = 224)	
Mean 24 h BP, mmHg	−0.019	reference	−0.003	0.003	0.027	0.561
Mean daytime BP, mmHg	−0.017	reference	−0.007	0.011	0.010	0.623
Mean nighttime BP, mmHg	−0.035	reference	0.027	−0.030	0.049	0.295
Mean 24 h BP, mmHg	−0.025	reference	0.005	0.008	0.029	0.491
Mean daytime BP, mmHg	−0.027	reference	0.008	0.011	0.009	0.453
Mean nighttime BP, mmHg	−0.019	reference	0.005	−0.013	0.063	0.594

suPAR = soluble urokinase plasminogen activator receptor; BMI = body mass index; SBP = systolic blood pressure; DBP = diastolic blood pressure; HbA1c = glycated hemoglobin 1Ac; LDL-C = low-density lipoprotein cholesterol; HDL-C = high-density lipoprotein cholesterol; hsCRP = high-sensitivity C-reactive protein; eGFR = glomerular filtration rate. ** *p* < 0.01; * *p* < 0.05.

**Table 4 diagnostics-13-02938-t004:** Baseline characteristics of the female study population according to suPAR quartiles.

Female Study Population	*n* = 1107				
Quartiles based on suPAR levels	Quartile 1 (*n* = 277)	Quartile 2 (*n* = 277)	Quartile 3 (*n* = 277)	Quartile 4 (*n* = 276)	*p*-Value
suPAR range (ng/L), women	≤1.30	1.30–1.60	1.60–2.00	≥2.00	
Age, years	37 (31–41)	37 (31–40)	36 (31–40)	37 (31–40)	0.720
BMI, kg/m^2^	23.0 ± 3.8	23.1 ± 3.6	23.1 ± 3.5	24.2 ± 4.1	<0.001
Regular physical activity, min/week	176 (64%)	183 (66%)	190 (69%)	184 (67%)	0.108
Irregular physical activity, min/week	101 (36%)	94 (34%)	87 (31%)	92 (33%)	
Alcohol consumption, g/day	0.0 (0.0–1.3)	0.0 (0.0–0.6)	0.0 (0.0–0.9)	0.0 (0.0–0.6)	0.176
Smoking (current), *n* (%)	38 (14%)	35 (19%)	49 (18%)	85 (31%)	
Smoking (past), *n* (%)	78 (28%)	67 (26%)	56 (20%)	50 (18%)	<0.001
Smoking (never), *n* (%)	161 (58%)	175 (56%)	172 (62%)	141 (51%)	
Mean office SBP, mmHg	113 ± 10	113 ± 10	114 ± 10	114 ± 11	0.784
Mean office DBP, mmHg	74.3 ± 8	75 ± 8	75 ± 8	75 ± 8	0.220
Mean arterial pressure, mmHg	90 ± 8	90 ± 8	91 ± 8	91 ± 9	0.406
Fasting glucose, mmol/L	4.6 ± 0.4	4.7 ± 0.4	4.7 ± 0.5	4.8 ± 0.5	0.005
HbA1c, %	5,3 ± 0.4	5.3 ± 0.3	5.4 ± 0.4	5.5 ± 0.4	<0.001
Cholesterol, mmol/L	4.9 ± 0.8	4.7 ± 0.8	4.7 ± 0.8	4.7 ± 0.8	0.001
LDL-C, mmol/L	2.8 ± 0.7	2.6 ± 0.7	2.7 ± 0.8	2.7 ± 0.7	0.858
HDL-C, mmol/L	1.8 ± 0.4	1.7 ± 0.4	1.7 ± 0.4	1.6 ± 0.4	<0.001
Triglyceride, mmol/L	0.7 (0.6–1.0)	0.7 (0.5–0.9)	0.7 (0.6–0.9)	0.8 (0.6–1.1)	0.283
hsCRP, mg/L	0.9 (0.5–2.0)	0.8 (0.4–1.9)	0.90 (0.5–1.8)	1.2 (0.6–2.8)	0.048
eGFR, mL/min/1.73 m^2^	141 ± 17	141 ± 18	140 ± 19	137 ± 16	0.009
Healthy lifestyle score	4.6 ± 1.1	4.6 ± 1.1	4.6 ± 1.1	4.3 ± 1.2	0.005
Framingham score	2.1 ± 1.1	2.1 ± 1.1	2.1 ± 1.1	2.1 ± 1.1	0.096
Subgroup (*n* = 1054)					
Quartiles based on suPAR levels	Quartile 1 (*n* = 264)	Quartile 2 (*n* = 264)	Quartile 3 (*n* = 263)	Quartile 4 (*n* = 263)	*p*-Value
Mean 24 h BP, mmHg	118 ± 10	116 ± 8	117 ± 9	118 ± 9	0.560
Mean daytime BP, mmHg	122 ± 10	120 ± 8	120 ± 9	121 ±10	0.562
Mean nighttime BP, mmHg	105 ± 11	104 ± 9	104 ± 9	105 ± 10	0.746
Mean 24 h BP, mmHg	76 ± 7	75 ± 6	75 ± 7	76 ± 7	0.732
Mean daytime BP, mmHg	79 ± 7	78 ± 7	78 ± 7	79 ± 7	0.824
Mean nighttime BP, mmHg	64 ± 7	64 ± 7	63 ± 6	64 ± 7	0.680

suPAR = soluble urokinase plasminogen activator receptor; BMI = body mass index; SBP = systolic blood pressure; DBP = diastolic blood pressure; HbA1c = glycated hemoglobin 1Ac; LDL-C = low-density lipoprotein cholesterol; HDL-C = high-density lipoprotein cholesterol; hsCRP = high-sensitivity C-reactive protein; eGFR = glomerular filtration rate.

**Table 5 diagnostics-13-02938-t005:** Multivariable linear regression analyses for the relationship between serum levels of suPAR and blood pressure parameters in women.

Female Study Population	*n* = 1107					
Quartiles Based on sST2 Levels	Continuous (*n* = 1107)	Quartile 1 (*n* = 277)	Quartile 2 (*n* = 277)	Quartile 3 (*n* = 277)	Quartile 4 (*n* = 276)	*p*-Value
BMI, kg/m^2^	0.031	reference	0.002	0.007	0.044	0.365
Irregular physical activity, %	0.053	Reference	0.013	0.062	−0.015	0.075
Alcohol consumption, g/day	−0.038	reference	−0.002	<0.001	−0.041	0.216
Current smoking, %	0.076	reference	−0.095 **	−0.014	0.144 **	0.017
Past smoking, %	−0.009	reference	−0.017	−0.037	−0.020	0.770
Mean office SBP, mmHg	0.014	reference	−0.004	0.005	−0.014	0.657
Mean office DBP, mmHg	0.060	reference	0.036	0.041	0.026	0.056
Mean Office MAP, mmHg	0.042	reference	0.018	0.025	0.008	0.186
Fasting glucose, mmol/L	0.011	reference	−0.012	0.026	0.049	0.718
HbA1c, %	0.081	reference	0.028	0.068	0.137 **	0.008
Cholesterol, mmol/L	−0.094	reference	−0.141 **	−0.120 **	−0.168 **	0.001
LDL-C, mmol/L	−0.051	reference	−0.075 *	−0.036	−0.063	0.102
HDL-C, mmol/L	−0.114	reference	−0.092 **	−0.123	−0.214	<0.001
Healthy lifestyle score	−0.037	reference	0.056	0.015	−0.104 **	0.226
Framingham score, points	0.034	reference	0.001	0.022	0.0034	0.261
Subgroup (*n* = 1054)						
Quartiles based on suPAR levels	Continuous (*n* = 1054)	Quartile 1 (*n* = 264)	Quartile 2 (*n* = 264)	Quartile 3 (*n* = 263)	Quartile 4 (*n* = 263)	*p*-Value
Mean 24 h BP, mmHg	−0.006	reference	0.052	−0.031	−0.026	0.849
Mean daytime BP, mmHg	−0.003	reference	0.054	−0.039	−0.026	0.915
Mean nighttime BP, mmHg	−0.003	reference	0.025	0.005	−0.026	0.909
Mean 24 h BP, mmHg	0.023	reference	0.022	−0.011	−0.008	0.439
Mean daytime BP, mmHg	0.025	reference	0.026	−0.027	−0.001	0.407
Mean nighttime BP, mmHg	0.024	reference	−0.009	−0.054	−0.038	0.424

suPAR = soluble urokinase plasminogen activator receptor; BMI = body mass index; SBP = systolic blood pressure; DBP = diastolic blood pressure; HbA1c = glycated hemoglobin 1Ac; LDL-C = low-density lipoprotein cholesterol; HDL-C = high-density lipoprotein cholesterol; hsCRP = high-sensitivity C-reactive protein; eGFR = glomerular filtration rate. ** *p* < 0.01; * *p* < 0.05.

## Data Availability

Anonymized data that underlie the results reported in this article are available upon justified request to the corresponding author.

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
