# Peer review of "The Association of suPAR with Cardiovascular Risk Factors in Young and Healthy Adults"

_diagnostics, 2023, doi:10.3390/diagnostics13182938_

Round 1

Reviewer 1 Report

The manuscript is an original study focused on the relationship between suPAR and cardiovascular risk factors and overall cardiovascular risk, respectively, in a young, healthy population.

The manuscript has many strengths and I will mention only the most significant:

The study methodology is rigorous, clearly presented and strictly adhered to.

The statistical analysis is thorough, demonstrating the authors' desire to provide only robust results. The results are numerous, but presented clearly and systematized.

Minor comments:

The Framingham risk score was developed based on an American population. There has been some criticism that the Framingham Risk Score has the potential to overestimate risk in European populations. As the current study enrolled a European population, it was expected that the SCORE risk prediction algorithm would be used. Improved algorithm (SCORE2) is now available. I think a brief comment on this choice would be of interest to readers.

The Discussion section could be improved with a paragraph focused on the pathophysiological background of the study findings.

In some tables, cholesterol is misspelled as cholesterin.

Thank you!

Author Response

A) The Framingham risk score was developed based on an American population. There has been some criticism that the Framingham Risk Score has the potential to overestimate risk in European populations. As the current study enrolled a European population, it was expected that the SCORE risk prediction algorithm would be used. Improved algorithm (SCORE2) is now available. I think a brief comment on this choice would be of interest to readers.

A) Thank you for your valuable thoughts and recommendations. Although the SCORE-2 is a valid alternative risk score, we used the Framingham score due to its broad applicability in estimating cardiovascular risk factors and  the available R codes.  

B)The Discussion section could be improved with a paragraph focused on the pathophysiological background of the study findings.

B) We agree and have now included the following explanations in the discussion:

B1) Association of smoking, HDL and cholesterol with suPAR:

“The biomarker suPAR is regarded as a marker of endothelial dysfunction and therefore also of atherosclerosis, which is why it seems plausible that smoking and higher cholesterol levels increase, whereas a higher HDL decreases plasma suPAR levels”

            B2) Association of BMI with suPAR:

The highest BMIs observed in the current study population (BMI of 38 kg/m2) are not classified as obesity grade III (BMI  40kg/m2), which was associated with higher suPAR plasma levels in the study published by Haupt, T.H. et al. [20]. The MONICA study resulted in the same conclusion as our study, that the BMI is not independently associated with suPAR plasma levels [23]. This conclusion could imply that comorbidities associated with obesity are causing higher plasma levels of suPAR, rather than a higher BMI itself.

            B3) Association of HbA1c with suPAR in female population:         

The sex-specific difference in this correlation can’t be explained by already existing literature, since a sex-independent correlation of suPAR and the endothelial dysfunction caused by permanently higher blood sugar levels reflected by the HbA1c levels would be expected.”

C) In some tables, cholesterol is misspelled as cholesterin. 

C) We are thankful for this advice and have corrected these typos accordingly.

Reviewer 2 Report

Review references, it's no necessary DOI, review format, please

Author Response

A)  Review references, it's no necessary DOI, review format, please

A) We thank you for this positive review and appreciate the comment about the references format that was adapted according.

Reviewer 3 Report

Wohlwend N. et al investigated the relation between soluble urokinase plasminogen activator receptor and cardiovascular risk factors and laboratory parameters including BMI, physical activity, alcohol consumption, smoking status, blood pressure parameters, glucose status, and lipid levels. Moreover, authors compared it to the healthy lifestyle score and the Framingham score representing the overall cardiovascular risk profile and found that suPAR plasma levels present strong correlations with specific cardiovascular risk factors with, however, sex specific differences.

Authors tried to present some interesting correlations between the variable they studied and cardiovascular risk factors with appropriate statistical methods. However, there are major flaws that need to be addressed. Specifically:

·     There are many correlations that cannot be explained by pathophysiology mechanisms.

·   I have doubts whether these findings could be implemented in clinical practice and be useful as diagnostic markers.

·   What are the future perspectives of your findings?

In general, English language is fine. Minor editing may be required in some points.

Author Response

A) There are many correlations that cannot be explained by pathophysiology mechanisms. 

A) We thank you for these helpful comment. We’ve now included multiple explanations for these correlations in the discussion:

A1) Association of smoking, HDL and cholesterol with suPAR:

The biomarker suPAR is regarded as a marker of endothelial dysfunction and therefore also of atherosclerosis, which is why it seems plausible that smoking and higher cholesterol levels increase, whereas a higher HDL decreases plasma suPAR levels”

      A2) Association of BMI with suPAR:

The highest BMIs observed in the current study population (BMI of 38 kg/m2) are not classified as obesity grade III (BMI  40kg/m2), which was associated with higher suPAR plasma levels in the study published by Haupt, T.H. et al. [20]. The MONICA study resulted in the same conclusion as our study, that the BMI is not independently associated with suPAR plasma levels [23]. This conclusion could imply that comorbidities associated with obesity are causing higher plasma levels of suPAR, rather than a higher BMI itself.

      A3) Association of HbA1c with suPAR in female population:    

“The sex-specific difference in this correlation can’t be explained by already existing literature, since a sex-independent correlation of suPAR and the endothelial dysfunction caused by permanently higher blood sugar levels reflected by the HbA1c levels would be expected.”

B) I have doubts whether these findings could be implemented in clinical practice and be useful as diagnostic markers. 

B) The goal of our study is to evaluate whether suPAR might act as a possible biomarker used to carry out individual risk stratification in order to allow preventive measures customized individually. Our study suggests that there are correlations of suPAR and the cardiovascular risk in men. On contrary, no such relations were found in female participants. In general, our findings are in contrast with previous studies, which primarily focused on older study populations. This led us to the conclusion, that “This is why we think that in elderly female participants with a higher prevalence of cardiovascular risk factors, such correlations might be found too. This suggests that suPAR might act as a predictor of the cardiovascular risk in elderly participants. “

C)What are the future perspectives of your findings?

C) We thank reviewer you for this comment. We’ve now added the following section to the discussion:

A possible approach for further investigations would be a follow-up study of equal study design, hypothesis and the same study population with the only difference of the study population being older in order to evaluate if suPAR might be an age-specific biomarker reflecting the cardiovascular risk solely in an elderly population. A second step would be to correlate individual changes in the plasma suPAR levels of each participant with the change in their individual cardiovascular risk profile.”

D) Required language editing:

D) Our manuscript went through the MDPI language and format editing before the initial submission.

Round 2

Reviewer 3 Report

Authors have answered all the reviewer comments sufficiently.